# Neural networks and brains share the gist but not the details

**Yingqi Rong (ryingqi@gmail.com)**
Department of Cognitive Science, Johns Hopkins University
3400 N Charles St
Baltimore, MD, USA 21218

**Colin Conwell (conwell@g.harvard.edu)**
Department of Cognitive Science, Johns Hopkins University
3400 N Charles St
Baltimore, MD, USA 21218

**Michael F. Bonner (mfbonner@jhu.edu)**
Department of Cognitive Science, Johns Hopkins University
3400 N Charles St
Baltimore, MD, USA 21218

## Abstract

Artificial neural networks (ANNs) excel in complex visual tasks and exhibit a notable alignment with neural representations in the ventral visual stream. Despite these advances, debates continue over whether ANNs fully capture the intricacies of the visual cortex. In this work, we explore test-time augmentation (TTA) as a novel strategy to enhance model-brain similarity without modifying model architectures. Traditionally employed to improve prediction accuracy through the averaging of augmented inputs, TTA here is leveraged to generate feature maps that more accurately predict neural responses to visual stimuli in the high-level ventral visual stream. We demonstrate that TTA consistently improves the prediction of neural activations across diverse architectures—including AlexNet, ResNet50, Vision Transformers, and robustified models—irrespective of their training datasets. Remarkably, averaging features from semantically similar but structurally varied augmentations, including those generated via diffusion models conditioned on text, can outperform representations derived from the original images. These results suggest that the conceptual gist, rather than detailed visual properties, underpins the enhanced model-brain alignment facilitated by TTA. We discuss potential mechanisms driving this effect and outline future directions for dissecting the interplay between augmented representations and neural processing in the visual cortex.

**Keywords:** test-time augmentation, visual cortex, fMRI, neural networks, encoding models

## Introduction

Artificial neural networks (ANNs) excel at complex visual tasks like object identification and image segmentation (Deng et al., 2009; Lin et al., 2014). Notably, ANNs outperform other computational models in aligning with brain representations of visual stimuli (Yamins et al., 2014; Yamins & Di-Carlo, 2016; Conwell, Prince, Kay, Alvarez, & Konkle, 2024). This alignment has inspired extensive research into identifying the optimal datasets, learning rules, architectures, and train-ing objectives for building the most "brain-like" models of vision (Schrimpf et al., 2018; Gokce & Schrimpf, 2024; Richards et al., 2019; Doerig et al., 2023).

Despite these efforts, debates persist over whether ANNs can fully model the visual cortex (Bowers et al., 2023; Golan et al., 2023; Linsley & Serre, 2023). Various findings highlight diverse approaches for improving model-brain alignment, such as using training on ecological data (Mehrer, Spoerer, Jones, Kriegeskorte, & Kietzmann, 2021), focusing on architectural inductive biases independent of large-scale pretraining (Kazemian, Elmoznino, & Bonner, 2024), or incorporating biological constraints like connectomic details (Margalit et al., 2024; Lappalainen et al., 2024).

In this paper, we explore a different approach that does not change the model itself, but nonetheless increases the ability of ANN representations to predict image-evoked cortical responses: test-time augmentation (TTA) of neural encoding models. Traditionally used in supervised learning to improve prediction accuracy, TTA involves making ensemble decisions based on multiple augmented versions of the original inputs (Krizhevsky, Sutskever, & Hinton, 2012; Szegedy et al., 2015; He, Zhang, Ren, & Sun, 2016; Kim, Kim, & Kim, 2020; Lyzhov, Molchanova, Ashukha, Molchanov, & Vetrov, 2020). For example, when a model categorizes an image, it doesn't just use the evidence (e.g., probability) of that single image, but the average sensory evidence based on multiple augmentations of the original image (e.g., rotated or cropped versions). This method can improve robustness and accuracy without requiring additional labeled data, additional training, or changes to the architecture. In the context of neural encoding models, we use TTA not to correct labels, but to improve the prediction of neural responses to images. Specifically, our objective is to map a network's representations to image-evoked fMRI responses in the ventral visual stream. However, rather than using the network's representations of the original images shown the fMRI subjects, we instead take the network's average response to a set of augmented images and use this to predict brain responses. Surprisingly, when

we use augmented images that are semantically similar to but structurally different from the original images (Rombach, Blattmann, Lorenz, Esser, & Ommer, 2022; Trabucco, Doherty, Gurinas, & Salakhutdinov, 2023), we find that ANN responses to these augmented inputs can yield better predictions of cortical representations than when using the original images that the subjects actually saw.

In follow-up analyses, we examined whether several key factors of a network's design are related to the effectiveness of TTA for neural prediction, and we compared multiple alternative approaches for data augmentation. First, we found similar benefits from TTA across multiple distinct architectures, including AlexNet, ResNet, and Vision Transformers (ViT), suggesting that these effects are not architecture dependent. Second, we found that TTA improved the predictions of robustified as well as conventional networks, suggesting that the effects of TTA are distinct from the effects of robustification. Third, we found that networks trained on less diverse datasets (such as faces) benefit the most from TTA, suggesting that TTA may help to overcome misalignment between the representations of a network and the representations of visual cortex. Finally, we found that the most effective approach for augmentation is to obtain the average response to a set of "semantic neighbor" images, which can be either synthesized conditioned on text descriptions or manually selected from a large pool of natural images. This procedure effectively coarse-grains the network's representations within a local semantic neighborhood. In contrast, conventional augmentations (e.g., cropping, rotation) and synthesis conditioned on images yielded substantially lower performance gains. In sum, these analyses suggest that the benefits of TTA for neural prediction can be observed across diverse networks and that these effects are specifically driven by improved alignment with the brain's representations of visual concepts rather than detailed image properties.

Together, our findings not only demonstrate a new approach for improving neural prediction through image augmentation and synthesis, they also have theoretical implications for understanding the nature of the shared representations between artificial neural networks and the ventral visual stream. Specifically, these findings suggest that these shared representations primarily reflect the coarse conceptual gist of images.

## Methods

### fMRI Dataset

The Natural Scenes Dataset (NSD) is a dataset that includes diverse natural stimuli and corresponding brain responses from eight subjects, collected via functional magnetic resonance imaging (fMRI) (Allen et al., 2022). Each subject viewed 10,000 different natural scene images across three different trials (Fig. 1A). Following the procedure in (Conwell et al., 2024), we analyze 1,000 visual stimuli seen by 4 subjects (1,2,5,7) and use their average neural responses across trials. We examined data from a large region of interest (ROI)

along the ventral visual stream, using the "ventral stream" ROI provided with the NSD dataset. Details about preprocessing, denoising and defining regions of interests(ROIs) are explained in the original NSD paper (Allen et al., 2022). The NSD images are a subset of the COCO dataset, cropped to a resolution of 425x425 pixels, and they include accompanying captions from five human subjects.

### Neural Network Models

For our analysis of architecture variation, we examined AlexNet, ResNet50 and ViT. We examined versions of these architectures pretrained on ImageNet as well as untrained versions. These networks were obtained from the Torchvision library. For our analysis of network robustness, we examined a robustified ResNet50 model pretrained on ImageNet (Engstrom, Ilyas, Salman, Santurkar, & Tsipras, 2019). For our analysis of training datasets, we examined instance-prototype contrastive learning (IPCL) models that were pretrained on datasets of faces, objects, places, or a mixture of these (Konkle & Alvarez, 2022). These IPCL models were downloaded from the Harvard Vision Lab (`https://github.com/harvard-visionlab/open_ipcl`). For all large networks including ResNet50 and ViT, we examined 10 layers sampled along the full depth of the network (i.e., relative depth from 0.1, 0.2, ..., 0.9 to 1).

### Encoding-Model Procedure

We followed the pipeline established in the DeepJuice toolbox developed by Conwell et al. (2024) to calculate voxelwise encoding scores for ANNs (Fig. 1B). We divided the image set into train and test sets with a fixed 500-500 split. For each ANN layer, we obtained activations to the full stimulus set and, for computational efficiency, reduced the dimensionality of the representations through sparse random projections, as per the Johnson–Lindenstrauss lemma (Li, Hastie, & Church, 2006). We then fit voxelwise encoding models in which a linear regression was used to map these ANN representations to the image-evoked fMRI responses. Specifically, we used ridge regression, with the optimal ridge penalty selected through leave-one-out cross-validation on the training set. We then applied the learned regression weights to the ANN representations for the held-out test images and generated predicted fMRI responses. We evaluated the accuracy of these encoding models by computing the Pearson correlation between the predicted and actual fMRI responses in the test set. We report the average encoding score across all voxels within the ventral stream ROI.

### Image Augmentation

For each stimulus image, we can use different types of augmentation or synthesis procedures to create image variations. For most of our analyses, we created 20 image variations by using a generative model,stable diffusion v1-5, to synthesize new images conditioned on human captions for the original image (Rombach et al., 2022). This procedure generates a set of semantic-neighbor images that are perceptually

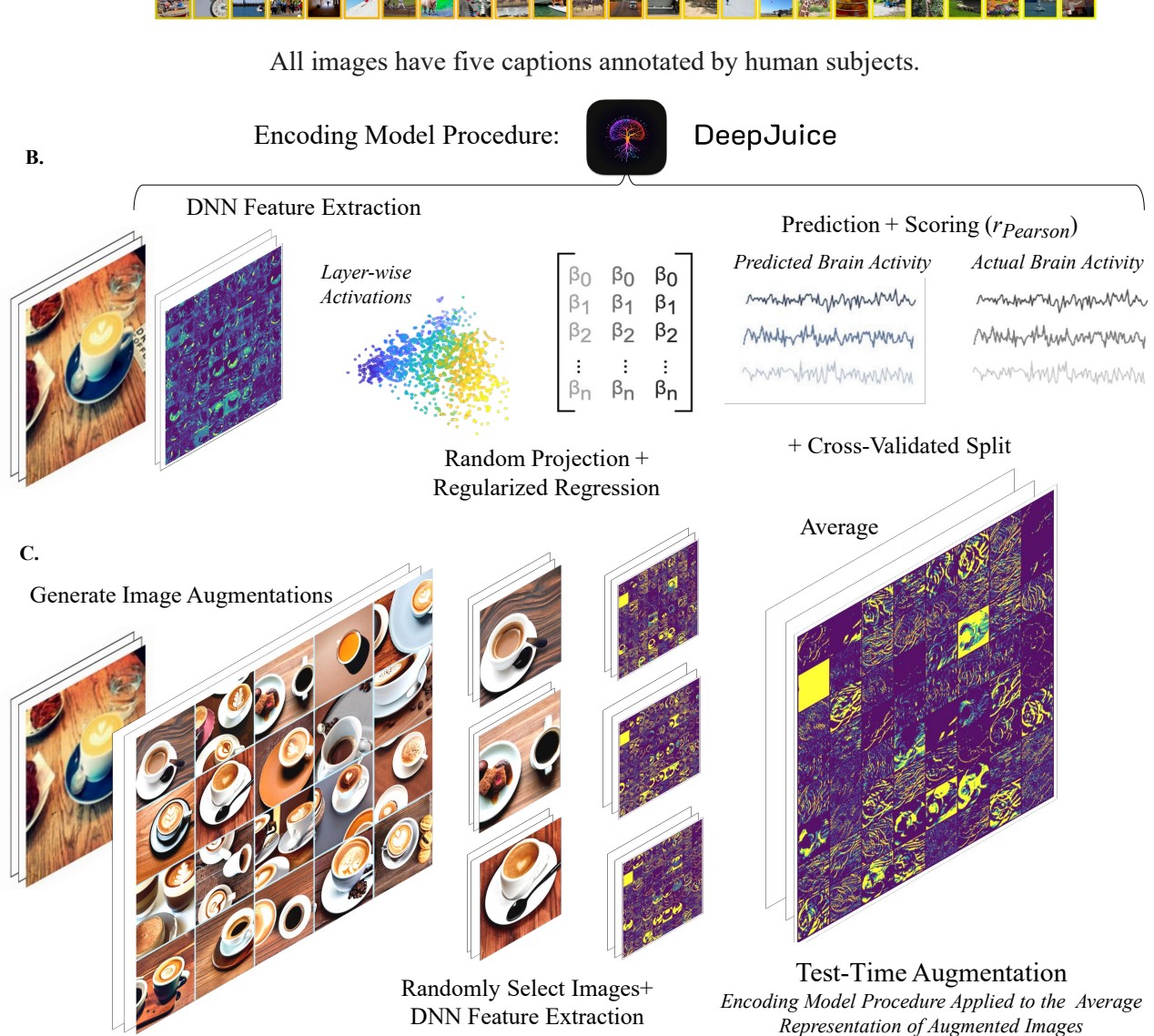

**Natural Scenes Dataset "Shared1000" Images**

**A.**

All images have five captions annotated by human subjects.

**B.** Encoding Model Procedure: DeepJuice

DNN Feature Extraction

Prediction + Scoring ($r_{Pearson}$)

*Predicted Brain Activity*    *Actual Brain Activity*

*Layer-wise Activations*

Random Projection + Regularized Regression

+ Cross-Validated Split

**C.** Generate Image Augmentations

Average

Randomly Select Images + DNN Feature Extraction

Test-Time Augmentation
*Encoding Model Procedure Applied to the Average Representation of Augmented Images*

Figure 1: A. We examine fMRI responses to the 1,000 images from NSD shared1000 stimulus set. Since the NSD stimuli were sourced from the COCO dataset, we have access to human-annotated captions for each image. B. This panels shows a visualization of the encoding-model procedure. Given any arbitrary neural network, feature maps are extracted for all images in each layer. These layer-wise feature maps undergo sparse random projection for dimension reduction and are then used in linear regression to predict fMRI responses to the images. Encoding models are fit on a set of training images and used to predict the response to held-out test images. C. This panel illustrates the basic procedure for test-time augmentation. Instead of obtaining a network's representation of the original stimulus, we instead obtain its average representation for an alternative set of augmented or synthesized images. This plot shows examples of synthesized images conditioned on text annotations. We pass the average network representation of the augmented/synthesized images as input to the encoding-model procedure.

distinct from the original image but semantically similar. We also examined the effectiveness of other augmentation procedures, including image synthesis conditioned on the original image rather than the caption (img2img), conventional auto-augmentation methods, and the selection of natural-image semantic neighbors. For auto-augmentation, we created 20 image variations with combinations of standard augmentations, including cropping, rotating, and recoloring (Cubuk, Zoph, Mané, Vasudevan, & Le, 2018). To select natural-image semantic neighbors, we sort other alternative images from the NSD dataset based on the cosine similarity of their annotation embeddings from CLIP (Radford et al., 2021). We take the first 20 nearest neighbors.

## Test-Time Augmentation

To apply TTA in our encoding-model procedure, we run the same encoding model analysis describe above, but instead of obtaining ANN activations to the original stimulus images, we obtained activations to an alternative set of augmented/synthesized images (Fig. 1C). For each original image, we randomly selected $k$ augmented images, with $k$ varied from 1 to 17. For $k > 1$, we computed the average ANN representation across the $k$ images. We repeated this procedure 20 times and obtained the average encoding performance across these repetitions.

## Results

We set out to determine how TTA affects encoding scores for ANN models of the ventral stream. Specifically, we examined layerwise encoding performance for ANN representations of the original stimulus images versus alternative sets of augmented/synthesized images. We performed these experiments on a variety of networks to explore the effects of TTA in different architectures, in trained versus untrained models, in a robustified network, and in networks trained with different datasets. We also examined a variety of approaches for performing TTA, including conventional auto-augmentations as well as novel approaches using image generation and ensembles of semantic neighbors.

## Test-Time Augmentation Through Text-to-Image Generation

We first examined TTA using the text-to-image synthesis approach, in which an alternative set of images were synthesized using a diffusion model conditioned on human-generated image descriptions. Examples of text-conditioned synthetic images are shown in Fig. 1C). These synthesized images are perceptually distinct from the original images but semantically similar. We examined encoding model performance when passing varied numbers of these synthesized images to the ANNs, instead of the original images.

***Pretrained networks with varied architectures.*** We first performed these analyses for three different architectures pretrained on ImageNet, namely AlexNet, ResNet50, and ViT. The results are shown in Fig. 2 (Figs. S1-4 show similar results in the dorsal stream). First, we can see that when

we replace each original-image representation with the representation of a single synthesized image, the encoding performance is always worse. This makes sense since we are essentially showing the ANN the wrong image (i.e., something other than the experimental stimulus). However, if we take the average representation of the ANN to multiple synthesized images, we find that encoding performance systematically increases with the number of synthesized images, and in many layers it even exceeds the performance for the original images (Fig. S5 confirms that there are no benefits from TTA when the synthesized images have a random association to the target image). This is a remarkable effect because it shows that in many layers of these ANNs, our predictions of image-evoked cortical responses can actually be *improved* by showing the ANN *different* images than what was actually shown to the human subjects in the fMRI experiment. These improvements were observed in all architectures, including classic convolutional architectures and a modern image transformer, and the improvements are generally higher in earlier network layers. The pronounced improvements in earlier layers are notable because these layers encode finer visual details. This suggests that the performance gains of TTA are due to averaging out the fine perceptual details that are idiosyncratic to individual images and, instead, emphasizing the visual concepts that are shared across a set of semantically similar images.

***Robustfied network.*** We wondered if similar improvements would also be observed if we applied TTA to a robustified network. We reasoned that the effectiveness of TTA could potentially be due to averaging out the kind of high-frequency information that typically makes networks vulnerable to adversarial attacks. However, contrary to this hypothesis, we found that TTA was just as effective when applied to a robustified ResNet (see the the bottom right panel of Fig. 2). This suggests that the effect of TTA is not due to the suppression of non-robust representations.

***Untrained networks.*** We also performed TTA analyses in untrained randomly initialized versions of AlexNet, ResNet50, and ViT (Fig. 3). The results are broadly consistent with those observed for the pretrained versions of these architectures, but in the untrained networks, performance gains are observed across the full depth of the networks, rather than being stronger in early layers. This difference in the layer-wise effects of trained and untrained networks may reflect the fact that in trained networks, higher layers already encode conceptual representations that are well-aligned with the brain and thus have less to gain from semantic-neighbor averaging. In contrast, the higher layers of untrained networks do not already encode conceptual representations and can thus obtain larger benefits from TTA.

***Effects of training data.*** We next sought to determine the effect of TTA on models trained with different classes of images (Fig. 4). Specifically, we examined networks trained on the same contrastive-learning objective but with categorically different training images, including faces, places, objects, and a mixture of these three. Again, we found that TTA improved

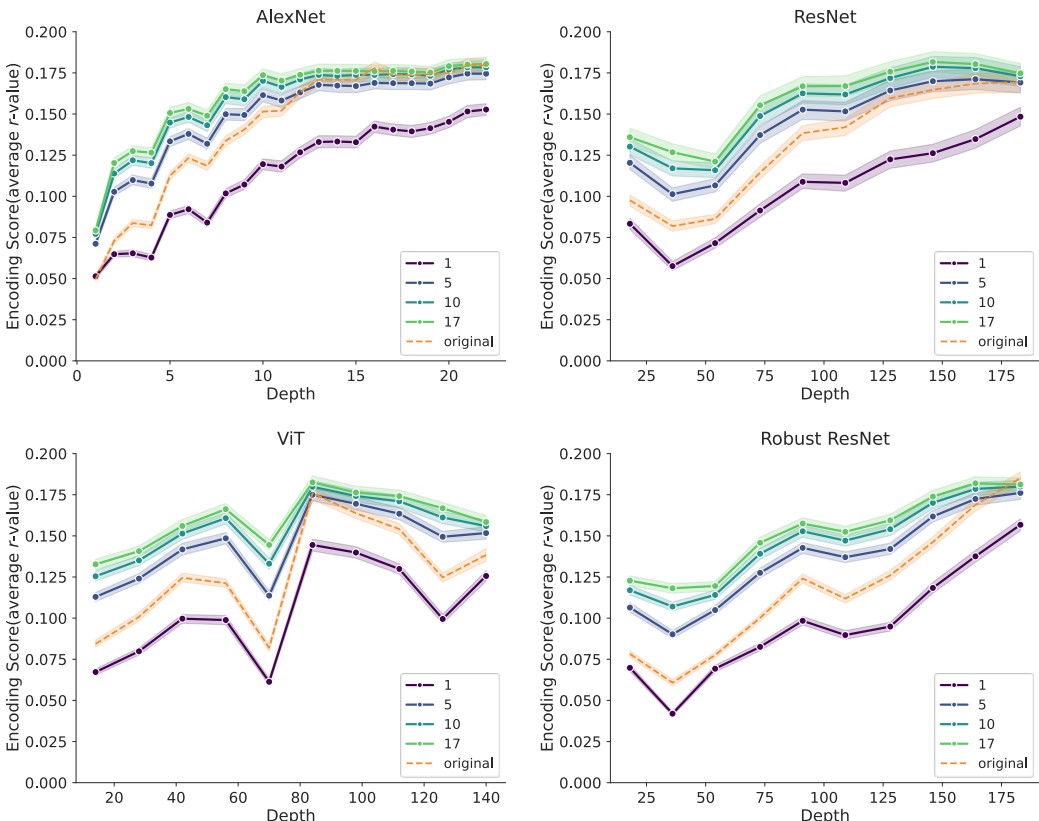

Figure 2: Effect of test-time augmentation in different networks using image synthesis conditioned on text descriptions. These line plots show the encoding scores for different layers along the depth of each network. Each panel shows the results for a different pretrained network, including AlexNet, ResNet50, ViT, and robustified ResNet50. The orange dashed line shows the encoding score obtained for the original stimulus images that were viewed by the subjects in the fMRI experiment. The other colored lines show encoding scores when using the network's responses to an alternative set of synthesized images, ranging from 1 to 17 synthesized images. The shaded bands show the standard deviation across subjects. These results show that in each network, there are many layers whose encoding scores are higher for the synthesized images than for the original images, with the encoding scores increasing as the number of synthesized images in increased. These effects are observed in distinct architectures, including both convolutional and transformer architectures, and in a robustified network.

the encoding performance, which was true for all four training datasets. Interestingly, in these networks, improvements were observed across all layers and in most cases, the largest improvements were observed in later layers. This suggests that for these networks, either the task objective or training sets (or both) yield weaker alignment with the conceptual representations of visual cortex, allowing for a greater benefit from TTA.

**Alternative Augmentation Strategies**

In all the preceding analyses, we used a novel TTA strategy in which we synthesized new images conditioned on text descriptions of the original images. The logic of this strategy is that by averaging the responses to these text-conditioned images, we are effectively coarse-graining the representations within a local semantic neighborhood and thus abstracting over perceptual details. However, we wondered if alternative augmentation procedures might be able to yield similar results. (See Fig. S6 for a summary plot allowing for direct

comparisons of augmentation strategies.)

***Conventional auto-augmentation.*** We first examined conventional auto-augmentation methods from the computer vision literature, which include cropping, rotating, and recoloring, among others (Fig. 5). Interestingly, this approach to TTA did not yield an improvement in encoding performance over the original images. This suggests that our approach of using semantic neighbors is more effective than the conventional augmentation procedures used in previous work. These results also demonstrate that the benefits of TTA can not be obtained by averaging any arbitrary image variations and, instead, require targeted manipulations.

***Image synthesis conditioned on the original images (img2img).*** We next examined TTA when using synthesized images that were conditioned on the original images rather than the text descriptions (Fig. 5). These synthesized images have clear variations relative to the original images, but they do not achieve the high degree of perceptual and struc-

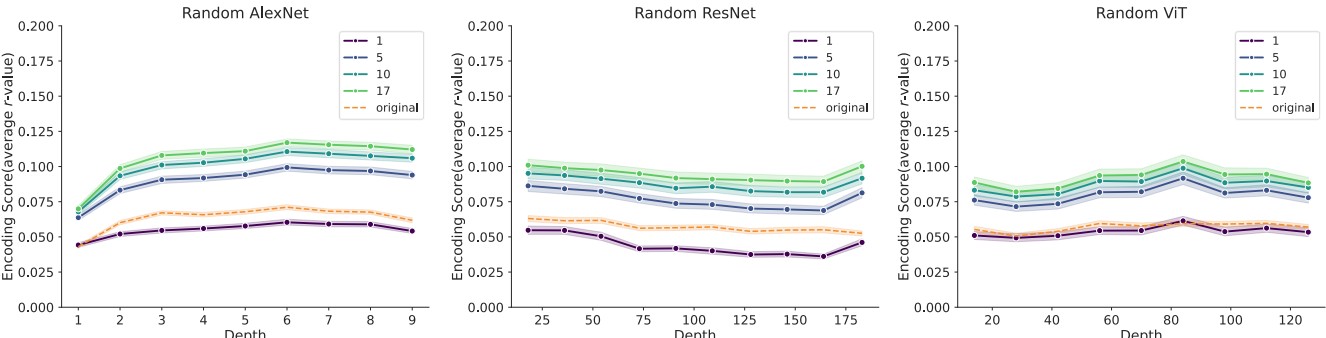

Figure 3: Effect of test-time augmentation in untrained networks using image synthesis conditioned on text descriptions. These line plots show the encoding scores for different layers along the depth of each network. Each panel shows the results for a different randomly initialized network, including AlexNet, ResNet50, and ViT. The orange dashed line shows the encoding score obtained for the original stimulus images that were viewed by the subjects in the fMRI experiment. The other colored lines show encoding scores when using the network's responses to an alternative set of synthesized images, ranging from 1 to 17 synthesized images. The shaded bands show the standard deviation across subjects. These results show that in each network and in all layers, the encoding scores are higher for the synthesized images than for the original images, with the encoding scores increasing as the number of synthesized images in increased. These effects are observed in distinct architectures, including both convolutional and transformer architectures.

tural variability that is obtained with text-conditioned synthesized images (Fig. 1C). This approach yielded little-to-no improvement in encoding performance over the original images, again suggesting that the use of perceptually diverse semantic neighbors is crucial.

**_Natural-image semantic neighbors._** Finally, we performed a version of the semantic-neighbor approach using natural images rather than synthesized images (Fig. 5). This approach yielded substantial improvements in encoding performance in early layers, resembling the patterns of improvement observed for the text-conditioned synthesized images shown in Fig. 2. This suggests that the encoding-score improvements from TTA depend on the use of semantic neighbors, which could be either synthesized or selected from a set of natural images.

## Discussion

We investigated whether ANN encoding models of the ventral stream could be improved through test-time augmentation (TTA) of their image inputs. Our findings show that the encoding performance of many ANN layers can be improved by using their average response to semantic-neighbor images rather than their response to the actual stimuli shown to the fMRI subjects. These effects were observed across multiple architectures, in both conventional and robustified networks, in trained and untrained networks, and in networks trained with different categories of images. The benefits of this TTA procedure were specifically due to the averaging of semantically related but structurally distinct images, and they were not observed when using conventional augmentation methods. Together, these findings demonstrate a new approach for improving the encoding performance of ANNs through tar-

geted augmentations of image inputs, and they suggest that the shared representations of ANNs and the ventral stream rely more on the conceptual content in images rather than their visual details.

As computational models become more accurate predictors of neural activity, they become more valuable tools for both basic research and translational applications. To this end, our work introduces TTA as a previously unexplored method that can benefit neuroscientists hoping to maximize the prediction accuracy of computational brain models (Schrimpf et al., 2018; Cichy et al., 2019). Moreover, TTA can serve not only as a technique for enhancing prediction accuracy but also as an analytical lens for dissecting the nature of shared representations between artificial and biological vision. For example, our findings challenge the basic idea that neural network models of visual cortex succeed by predicting how the brain responds to the specific perceptual content in an image. Instead, they suggest that in some cases, neural networks and the brain may diverge in how they represent the details of an image but agree on how they represent an image's latent visual concepts. More broadly our work suggests that a fruitful approach for probing the nature of the shared representations between neural networks and the brain is through the systematic manipulation of an encoding model's inputs, which can target either low-level perceptual properties through image augmentations or high-level conceptual properties through image generation.

The most common approaches for image augmentation involve simple image transformations, like crops, rotations, and color changes (Krizhevsky et al., 2012). While these augmentations change the low-level properties of the images, they do not induce structural changes to the image content. We used

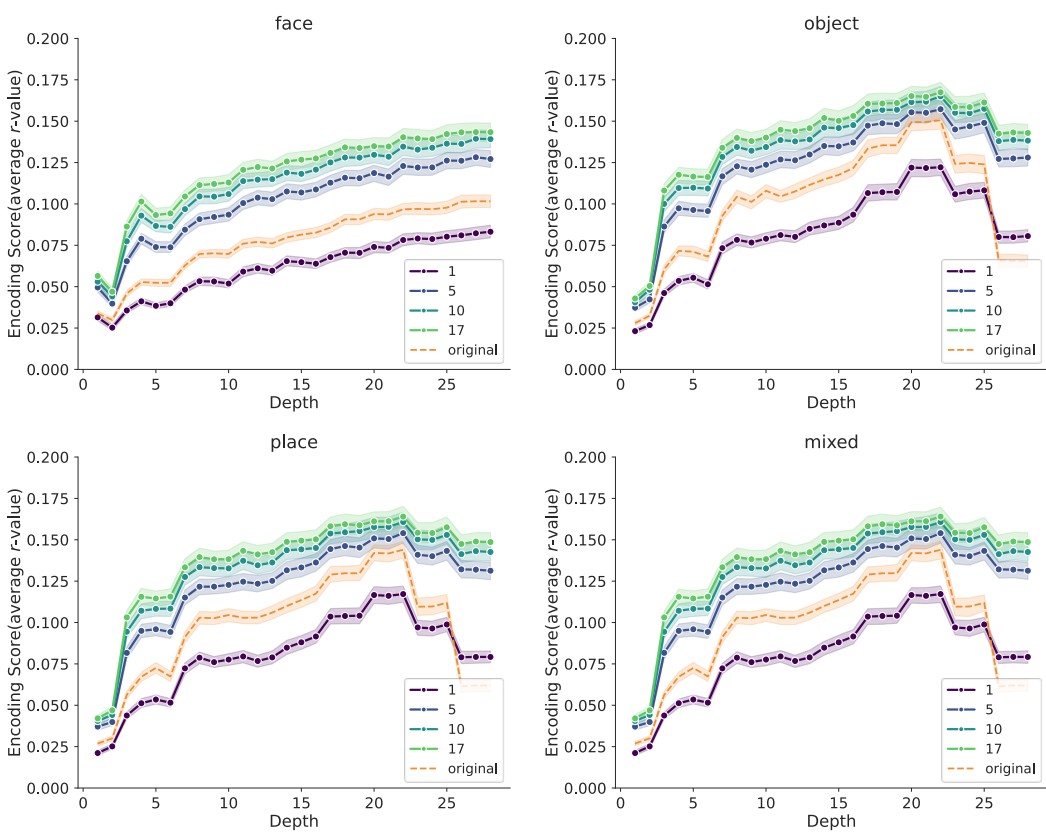

Figure 4: Effect of test-time augmentation in networks with varied training data using image synthesis conditioned on text descriptions. These line plots show the encoding scores for different layers along the depth of each network. Each panel shows the results for a contrastive-learning models trained with categorically different images, including faces, places, objects, and a mixture of these three. The orange dashed line shows the encoding score obtained for the original stimulus images that were viewed by the subjects in the fMRI experiment. The other colored lines show encoding scores when using the network's responses to an alternative set of synthesized images, ranging from 1 to 17 synthesized images. The shaded bands show the standard deviation across subjects. These results show that in each network and in all layers, the encoding scores are higher for the synthesized images than for the original images, with the encoding scores increasing as the number of synthesized images in increased.

image synthesis conditioned on text descriptions to sample structurally diverse images that are nonetheless semantically similar. This synthesis-based approach proved to be markedly more effective than conventional augmentations. Moreover, we found that a related procedure based on the selection of natural-image semantic neighbors was similarly effective. This suggests that the benefits of this TTA procedure stem from its ability to emphasize core conceptual features while averaging out idiosyncratic visual details.

The importance of semantic-neighbor averaging suggests that many ANN layers share coarse conceptual representations with the ventral stream but differ from the ventral stream in how they represent detailed perceptual content. This is somewhat surprising given that the visual ANNs examined here are feedforward models trained only on static images and would thus be expected to primarily represent perceptual content. Despite the primacy of perceptual information in these ANN representations, our findings suggest that their

latent conceptual representations may be doing much of the work in predicting ventral stream representations to natural images. This idea makes connections to other recent work showing that ventral stream representations can be predicted surprisingly well by language model representations based on image descriptions (Wang, Kay, Naselaris, Tarr, & Wehbe, 2023; Doerig et al., 2022)—an effect that appears to be driven by descriptions of objects and other visual content (Conwell, Prince, Alvarez, & Konkle, 2023; Shoham, Broday-Dvir, Malach, & Yovel, 2024). (See Fig. S7 for the performance of language embeddings in our analyses.) In line with this previous work, our findings suggest the intriguing possibility that the lingua franca of visual cortex and ANNs is grounded in visual concepts rather than the perceptual details of images. This would explain the striking performance of both language-based representations and semantic-neighbor averaging in ANN-based models of visual cortex.

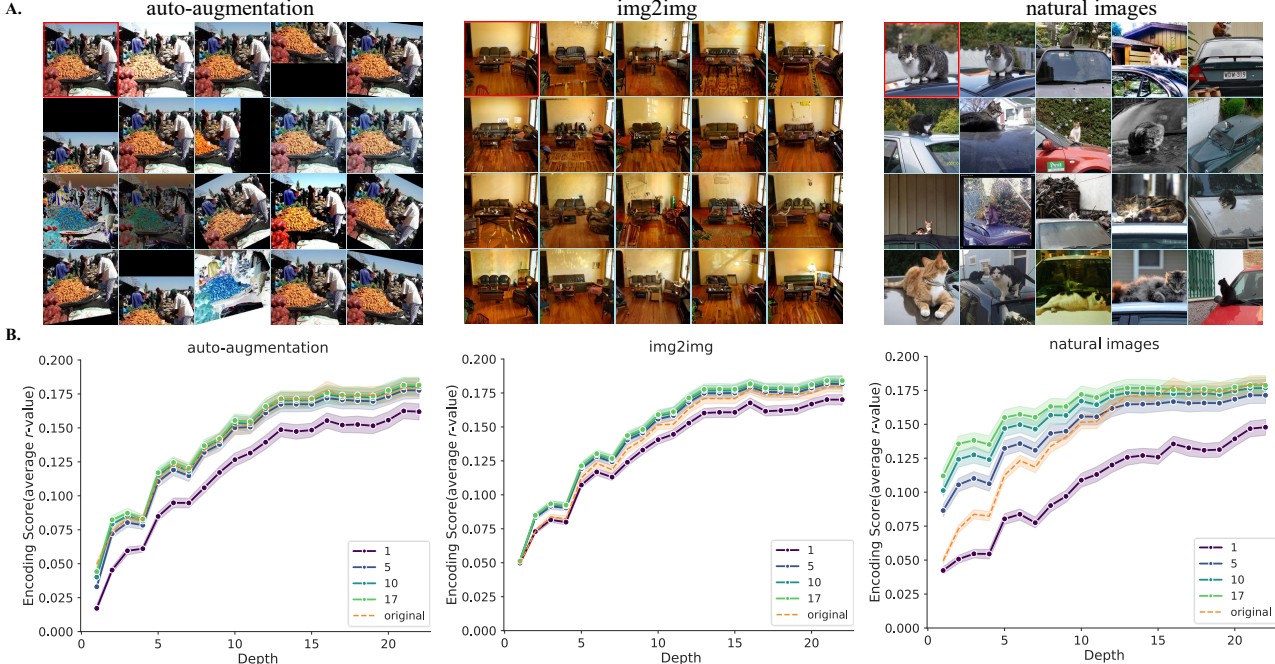

Figure 5: Effect of test-time augmentation when using alternative augmentation strategies. A. This panel shows samples of images obtained using three different augmentation procedures: auto-augmentation, synthesis conditioned on the original image (here labeled img2img), and natural-image semantic neighbors (here labeled natural images). B. These line plots show the encoding scores for different layers along the depth of the network. These analyses show the performance of AlexNet pretrained in ImageNet. The orange dashed line shows the encoding score obtained for the original stimulus images that were viewed by the subjects in the fMRI experiment. The other colored lines show encoding scores when using the network's responses to an alternative set of synthesized images, ranging from 1 to 17 synthesized images. The shaded bands show the standard deviation across subjects. These results show that auto-augmentation and synthesis conditioned yield little-to-no performance gains over the original images. However, the use of natural-image semantic neighbors yields substantial improvements, specifically in early layers.

## Code

Our codebase is publicly available on GitHub at https://github.com/BonnerLab/diffuse-encoder. It includes both a demo that illustrates the overall workflow and a complete implementation of our method.

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

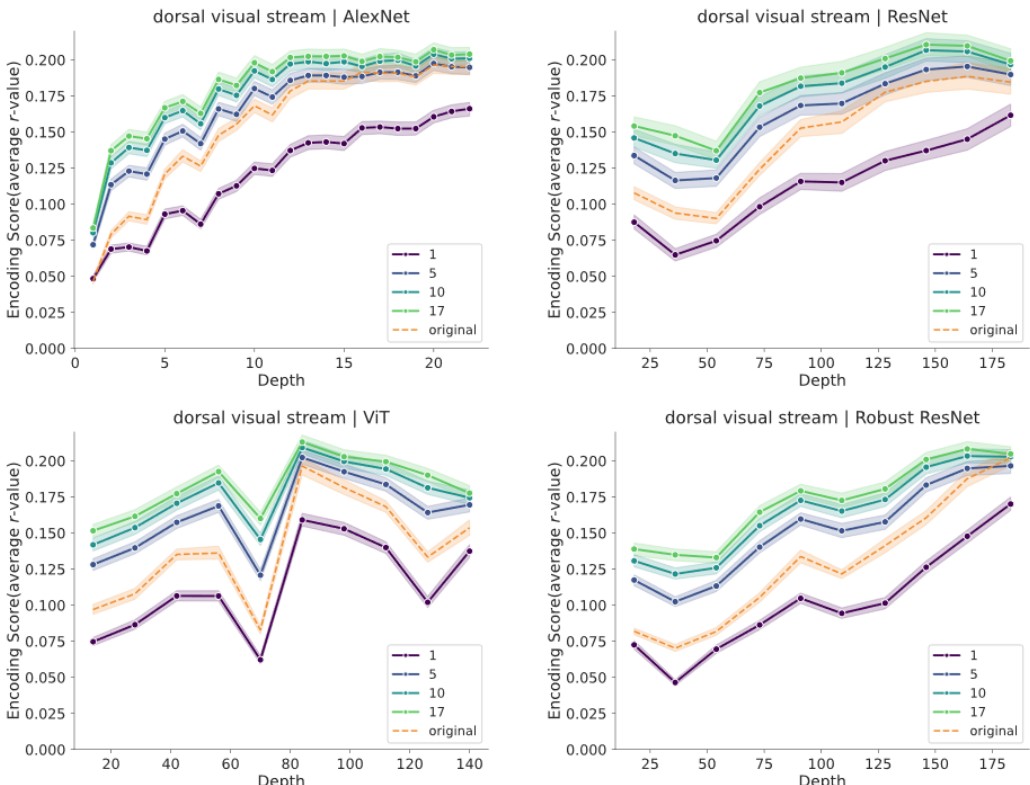

**Figure S1:** Effect of test-time augmentation in different networks using image synthesis conditioned on text descriptions in the dorsal stream. These line plots show the encoding scores for different layers along the depth of each network. Each panel shows the results for a different pretrained network, including AlexNet, ResNet50, ViT, and robustified ResNet50. The orange dashed line shows the encoding score obtained for the original stimulus images that were viewed by the subjects in the fMRI experiment. The other colored lines show encoding scores when using the network's responses to an alternative set of synthesized images, ranging from 1 to 17 synthesized images. The shaded bands show the standard deviation across subjects. These results show that in each network, there are many layers whose encoding scores are higher for the synthesized images than for the original images, with the encoding scores increasing as the number of synthesized images in increased. These effects are observed in distinct architectures, including both convolutional and transformer architectures, and in a robustified network. Results are for the parietal stream region as defined in the NSD dataset.

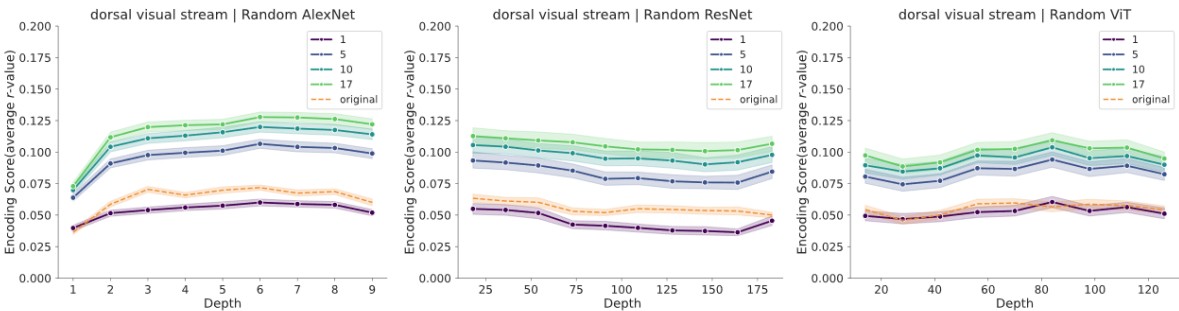

**Figure S2:** Effect of test-time augmentation in untrained networks using image synthesis conditioned on text descriptions in the dorsal stream. These line plots show the encoding scores for different layers along the depth of each network. Each panel shows the results for a different randomly initialized network, including AlexNet, ResNet50, and ViT. The orange dashed line shows the encoding score obtained for the original stimulus images that were viewed by the subjects in the fMRI experiment. The other colored lines show encoding scores when using the network's responses to an alternative set of synthesized images, ranging from 1 to 17 synthesized images. The shaded bands show the standard deviation across subjects. These results show that in each network and in all layers, the encoding scores are higher for the synthesized images than for the original images, with the encoding scores increasing as the number of synthesized images in increased. These effects are observed in distinct architectures, including both convolutional and transformer architectures. Results are for the parietal stream region as defined in the NSD dataset.

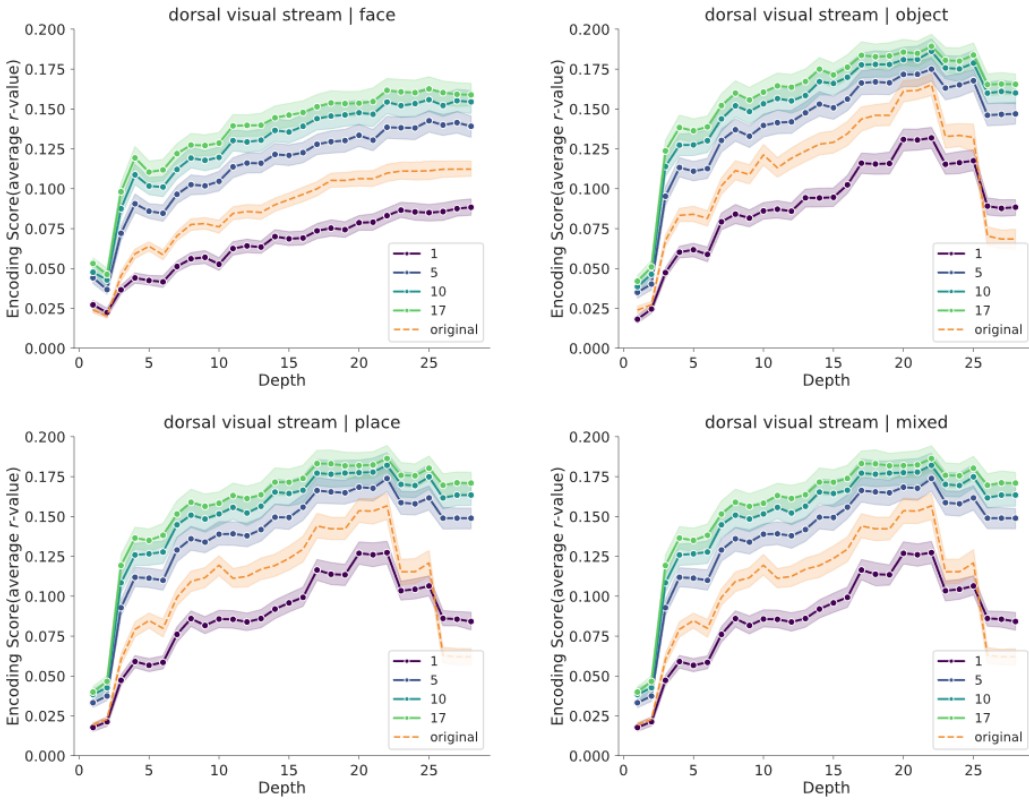

**Figure S3:** Effect of test-time augmentation in networks with varied training data using image synthesis conditioned on text descriptions in the dorsal stream. These line plots show the encoding scores for different layers along the depth of each network. Each panel shows the results for a contrastive-learning models trained with categorically different images, including faces, places, objects, and a mixture of these three. The orange dashed line shows the encoding score obtained for the original stimulus images that were viewed by the subjects in the fMRI experiment. The other colored lines show encoding scores when using the network's responses to an alternative set of synthesized images, ranging from 1 to 17 synthesized images. The shaded bands show the standard deviation across subjects. These results show that in each network and in all layers, the encoding scores are higher for the synthesized images than for the original images, with the encoding scores increasing as the number of synthesized images in increased. Results are for the parietal stream region as defined in the NSD dataset.

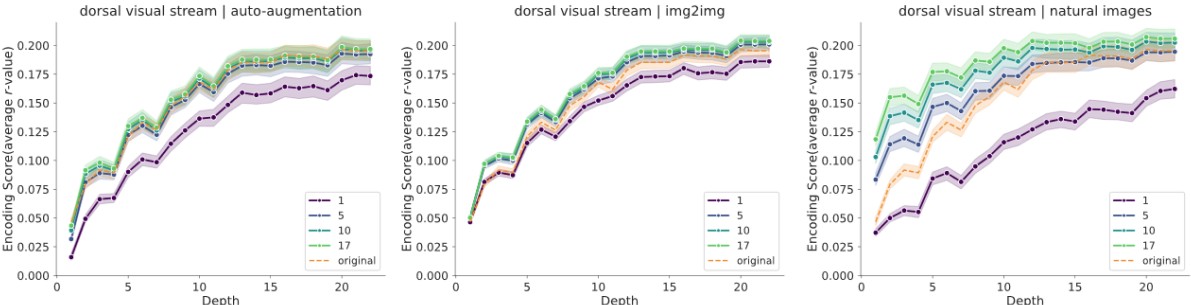

**Figure S4:** Effect of test-time augmentation when using alternative augmentation strategies in the dorsal stream. These line plots show the encoding scores for different layers along the depth of the network. These analyses show the performance of AlexNet pretrained in ImageNet. The orange dashed line shows the encoding score obtained for the original stimulus images that were viewed by the subjects in the fMRI experiment. The other colored lines show encoding scores when using the network's responses to an alternative set of synthesized images, ranging from 1 to 17 synthesized images. The shaded bands show the standard deviation across subjects. These results show that auto-augmentation and synthesis conditioned yield little-to-no performance gains over the original images. However, the use of natural-image semantic neighbors yields substantial improvements, specifically in early layers. Results are for the parietal stream region as defined in the NSD dataset.

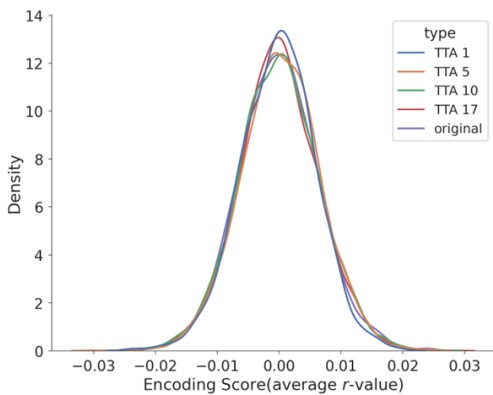

**Figure S5:** Empirical null distributions show that there are no benefits of TTA when the image variations have a random association to the target image. We randomly permuted the original images or sets of semantic-neighbor images used to generate predicted responses in the test set. This procedure was repeated 1,000 times to generate null distributions. As expected, the null distributions are centered at zero and look similar for analyses involving original images and TTA image sets, demonstrating that averaging responses to randomly associated images does not improve performance.

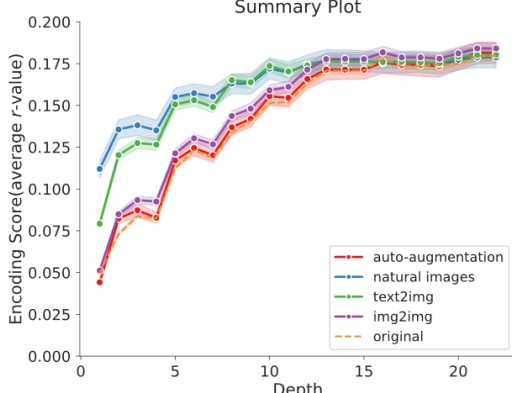

**Figure S6:** This plot shows the encoding scores obtained from different TTA procedures applied to layers along the depth of AlexNet. The orange dashed line shows the encoding score obtained for the original stimulus images that were viewed by the subjects in the fMRI experiment. The other colored lines show encoding scores obtained when using the network's responses to an alternative set of 17 images, which were created using different augmentation strategies. The shaded bands show the standard deviation across subjects.

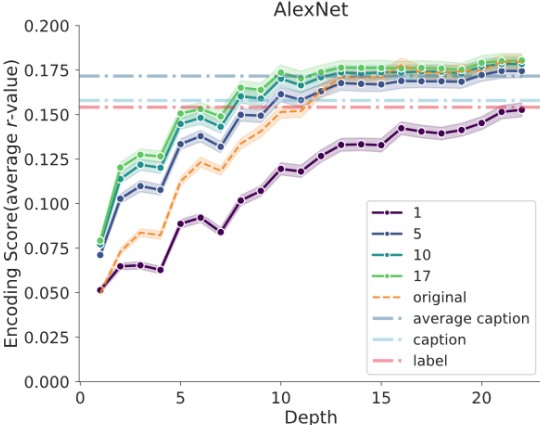

**Figure S7:** This plot shows the encoding score for different layers along the depth of AlexNet. The orange dashed line shows the encoding score obtained for the original stimulus images that were viewed by the subjects in the fMRI experiment. The other colored lines show encoding scores when using the network's responses to an alternative set of synthesized images, ranging from 1 to 17 synthesized images. The shaded bands show the standard deviation across subjects. The dark blue dashed line shows the average encoding score obtained when using the average CLIP caption embedding (average of four captions randomly subsampled from five captions five times). The light blue dashed line shows the average encoding score obtained when using individual caption embeddings (average of the scores for five individual captions). The red dashed line shows the encoding score for the embedding of the concatenated object and scene labels associated with each scene.

