# OpenReview forum: "Neural networks and brains share the gist but not the details"
_ccneuro.org/CCN/2025/Proceedings — CCN 2025 Proceedings asProceedingsPoster_

### Official Review · Reviewer_tFTC · 2025-03-31
**An interesting paper**

**Soundness:** 1
**Clarity:** 2

**Comments:**

Interest:
The whole paper is pretty interesting.
However, the authors could further elaborate on the motivation for improving the model's prediction performance of image-evoked human cortical responses.
Additionally, the current results seem to offer limited insights for cognitive science or neuroscience.

Soundness:
I am concerned that the results may not entirely align with the author's intended interpretation. If the use of perceptually diverse semantic neighbors is indeed critical, wouldn't we expect to see improvements in fMRI prediction performance from the models' later layers? However, the findings instead show enhanced fMRI prediction based on models' early layers, which are primarily involved in encoding low-level visual features.

Clarity:
1. Could the authors also provide the results for different brain regions?
2. The authors should conduct additional control experiments - trained encoding models based on randomly selectived images - can the encoding model trained on averaging model activations across multiple (more than one) images but not visually or semantically similar to each other still perform higher fMRI prediction performance?

Some other issues:
1. Why did the authors only select 4 subjects from NSD?
2. Why did the authors only use the shared 1000 visual stimuli? There are many more images and the corresponding fMRI responses in NSD, and they can test the model on the shared 1000.
3. How did the author make the fixed 500-500 split?
4. Why did the results on untrained models show improvements across all layers of the network? It needs more explanations.
5. Why did the models trained on face, object, place, mixed datasets show improvements not only on early but also late layers of the network? It needs more explanations.
6. It would be better to have a figure combining the criticial results based on different augumentation strategies. I would like to see the comparisons among multiple strategies.

**Expertise:**

3

**Interest:**

2

---

> ### Author Rebuttal · Authors · 2025-04-13
>
> Thank you for the thoughtful and constructive feedback. In response, we have done the following:
>
> 1. Added a new paragraph to the Discussion to better explain the motivation and implications of TTA for neuroscientists (all edits are in blue text).
> 2. Added text to the Results on the interpretation of effects in early layers. Note that below, we also provide a detailed response.
> 3. Performed analyses in a dorsal stream ROI to show that our findings extend beyond the ventral-stream region used in the main figures (see Figs. S1-4).
> 4. Reported empirical null distributions to show that there is no benefit of TTA when the images have a random association to the target image (see Fig. S5).
> 5. Added text to the Results to explain our interpretation of the improvement across all model layers in untrained networks and in networks with varied-training sets.
> 6. Added a supplementary figure with different TTA methods (see Fig. S6).
>
> There was one comment on the Soundness of the paper, which had to do with our interpretation of the layer-wise effects. Thank you for raising this question. We realized that we had never explicitly explained how this pattern of results aligns with our interpretation. Importantly, we want to clarify that the greater benefit of TTA in early layers is indeed consistent with our semantic neighborhood interpretation. The reason for this is that early layers encode fine details of the image, and it is precisely these lower-level details that are misaligned with the brain. When we apply TTA with semantically similar but visually diverse inputs, we effectively average out idiosyncratic visual details that may be represented differently in networks versus brains. What remains after this averaging are more conceptual aspects of the representation that are common across a set of semantic-neighbor images. In later layers, networks have already abstracted away from these idiosyncratic details toward more stable conceptual representations that better align with the brain, leaving less room for improvement through TTA.
>
> Our analysis of NSD follows the procedures from Conwell et al. 2024, which were designed to address the computational demands of performing high-throughput encoding model analyses. We faced the same challenge for our analyses, which involved multiple networks, many layers, and many sets of images (e.g., 20 times the number of original images for each TTA procedure).

---

### Official Review · Reviewer_CNaf · 2025-03-31
**Review of "Test-Time Augmentation for Model-Brain Similarity"**

**Soundness:** 3
**Clarity:** 3

**Comments:**

This paper uses a novel data augmentation technique to improve the correspondence between deep neural network models of visual recognition and neural responses in the ventral cortex to natural scenes. The findings are likely to be of great interest to the CCN community, as they revolve around understanding why high-dimensional embeddings in deep neural networks are such a remarkably good predictor of ventral cortex activity. These findings are also both counterintuitive and novel, asking the field to reconsider its assumptions about the main factors that lead to a higher or lower correspondence between a neural response and a model’s response to a natural scene.

The strengths of the paper are many: I appreciate that they use a large number of models to validate their findings, (2) the figures are informative and illustrate the findings well, and (3) the authors conduct several quality checks on their findings, showing robustness across different model architectures, trained vs. untrained models, and (3) make explicit comparisons with other kinds of data augmentation techniques.

That being said, I have some suggestions for improvement, especially with respect to interpretation and the terminology that is being used. I’m not sure that I am confident that these results are being driven by anything that is “abstract” or “semantic”. What is meant by the brain’s “abstract semantic representation of images”?  I think this needs a little unpacking and connecting to previous literature in order to avoid misinterpretation.  In one sense, “abstract” can refer to something that is divorced or separable from perceptual input—this is also true of the word “semantic”, which can often be taken to mean the similarity structure that isn’t necessarily reflected in the structure of the visual world (e.g., the fact that beluga whales are mammals is a semantic distinction that isn’t obvious from perception alone).   Yet of course the perceptual structure of the world is naturally correlated with the semantic structure of the world in meaningful ways—objects occur in predicative locations/scenes, categories (e.g., coffee cups, as in their example) are characterized by meaningful sets of perceptual features that relate to their function (e.g., can hold liquid). My guess is that the paper doesn’t mean to imply that ventral cortex (or these models) are necessarily representing concepts that are separable from perceptual inputs: instead, I take them as providing evidence that ventral cortex is coding for “visual concepts” or meaningful perceptual units that extend beyond the exact colors, curvature, and orientation statistics of the images that participants are shown in the images. That is still a significant and interesting finding, if true—but this is somewhat different than saying that deep neural networks and ventral cortex primarily represent the “abstract semantic” properties of objects.

I’d also like to see their findings connected to recent, adjacent work— I know much of this is unpublished, but this field is moving at a rapid pace! Conwell et. al., 2023 suggests that the reason that embeddings LLMs are “unreasonably effective” at predicting activity in the ventral cortex is because they capture information about which objects are present in the image. Other recent work (Shoham et al., 2024) suggest that abstract text descriptions are not effective predictors of neural activity in their EEG paradigm. I should say I am not one of these authors, and have not gone into detail in these papers, but I was struggling to put all of these recent findings together given their differences in terminology.

Perhaps one analysis in my mind that is missing—if they authors want to engage in this debate, which it appears they do given their conclusion statements—is a direct comparison to linguistic embeddings for the captions of the images  (these could easily be obtained using the same CLIP architecture that is used at other points in the paper for the semantic neighbors). Presumably, these captions could also be “augmented” in the same fashion as the visual tokens, or reduced to the object tokens that are present in the image (for comparison to prior literature). The degree to which embeddings from captions, object labels, or augmented versions of those descriptors capture the same variance in neural responses would be informative.

Relatedly, the paper describes results where TTA has a greater or weaker effect for earlier vs. later model layers, but doesn’t spend much time elaborating on why or interpreting these differences. Of course, this is related to the larger point above.  I think that the layer-wise analyses merit more consideration and connection to the broader argument. If these results do reflect “abstract semantic alignment”, I think I would have predicted—before seeing the results—a weaker or non-existent effect in earlier layers, and a pronounced effect in later layers. I can’t quite work out the logic of what the authors expected here, or perhaps they didn’t have a strong expectation (but it isn’t discussed in the GD, despite being part of almost every major figure).

Most of this sounds quite negative, so I should highlight that overall I thought this was a creative and important paper that is trying to advance our understanding of why we see a correspondence between ANNs and neural representations. I understand that the text-based analysis might be out of scope for a tight timeline on a revision, but I’d like the authors to expand on their predictions and interpretations of the layer-based analyses, and clarify their terminology and predictions.


Minor:
1. Why did the authors stop at 17 images? Is there a technical or computational reason?
2. The title only connects with scientists who happen to be familiar with TTA before this paper, rather than describing the main finding
3. Upon publication, I would advocate for the authors to publish their codebase. This will help build towards a cumulative science and help resolve discrepancies with previous findings

**Expertise:**

3

**Interest:**

3

---

> ### Author Rebuttal · Authors · 2025-04-13
>
> Thank you for this detailed and helpful feedback. Based on these comments, we have done the following:
>
> 1. Edited terminology throughout the manuscript to be more precise about visual concepts versus abstract semantics (all edits are in blue text).
>
> 2. Addressed the suggested relevant literature in the Discussion.
>
> 3. Performed the suggested encoding model analyses using CLIP embeddings (see Fig. S7).
>
> 4. Added text to the Results section on the interpretation of layer-wise effects. Note that we also explain how we interpret the benefits of TTA in early layers in our response to Reviewer 3.
>
> 5. Updated the title to be more familiar and engaging to an audience of neuroscientists.
>
> Additionally, we would like to note the following:
>
> • We explain the choice of 17 images in our response to Reviewer 1.
>
> • We are currently working on a code repo that we plan to share at the time of publication.

---

> > ### Comment · Reviewer_CNaf · 2025-04-21
> >
> > Thank you for engaging with all aspects of my review. I feel that the new version is significantly improved and I now feel that this paper makes a substantive contribution to the CCN community that advances our understanding of model--brain similarity metrics. I appreciate the additional analyses conducted on a tight timeline--both in response to my own concerns as well as that of other reviewers--as well as the revisions to the writing. Bravo!
> >
> > I am updating my review scores and look forward to discussing this paper at CCN.

---

### Official Review · Reviewer_7qbB · 2025-04-01
**An interesting toolkit but I am not sure why TTA works**

**Soundness:** 3
**Clarity:** 2

**Comments:**

The paper is of broad interest and is clearly written. Their conclusions seem justified as well. I do feel that experts in the field might not always be ready to accept the usage of different stimuli for the models and the subjects.

Questions:

Why motivated the choice of k = 17 in Fig 2?

Fig 3: Why is k = 1 worse than the original (orange line) for Untrained AlexNet and ResNet? Why does it matter to an untrained network if the wrong image was shown? If the difference is not significant, please include significance testing results. Furthermore, the legend occludes two of the lines in the plot, which can be avoided by moving the legend to the top left.

Has it been verified that the image generation models don't result in any artifacts? I know older generative models had issues with artifacts, but given how good modern generative models are, I wouldn't be surprised if they don't have this issue (but I don't know the modern gen AI literature well enough to be sure).

Figure 4 doesn't seem to be referred in the text.

The paper is interesting but I am still uncertain why TTA helps at all. To the authors' credit, they have tried various factors but it does not seem like we know why TTA helps yet.

**Expertise:**

2

**Interest:**

3

---

> ### Author Rebuttal · Authors · 2025-04-13
>
> Thank you for this thoughtful and constructive feedback. Below are responses to each of the five questions:
>
> 1. We found that there are diminishing returns when doing TTA with more than ~10-20 image variations. This is why synthesized/selected 20 alternate images for each image in the fMRI dataset. We then wanted to perform the analyses with multiple random subsamples of these 20 images, which is why we set the max at 17 images. The details of this random subsampling procedure are explained in the Test Time Augmentation subsection of the Methods.
>
> 2. We see a similar trend in the trained networks. The underlying explanation is the same for both trained and untrained networks. Importantly, both trained and untrained networks encode image features that are useful for predicting image-evoked cortical responses (our findings and many previous studies show such effects for untrained networks). However, when these networks are shown the “wrong” image, their feature representations will also be wrong, and as a result, they will be less predictive of cortical responses. Also, we have updated the figure to address the issue with the legend.
>
> 3. We are not sure what specific kinds of artifacts you are referring to. However, it is true that sometimes the images have elements that seem unnatural (e.g., weird textures, weird shapes). Nonetheless, this doesn’t seem to be a problem for TTA. Also, we want to emphasize that we get similar results when performing TTA with natural-image semantic neighbors.
>
> 4. Figure 4 is referred to in the subsection of the Results titled “Effects of training data.”
>
> 5. We agree that the benefits of TTA are not easily explained based on common assumptions about how DNNs predict brain representations. We speculate in the Discussion about potential explanations based on the idea that DNNs and brains may not be well-aligned in how they represent the detailed properties of images. However, regardless of this explanation, we believe that our findings are relevant to the community precisely because they challenge common assumptions and raise intriguing new questions that might bring us to a better understanding of the shared representations between brains and DNNs.

---

> > ### Comment · Reviewer_7qbB · 2025-04-20
> >
> > I am satisfied with the authors' response and maintain my recommendation to accept the paper. I do like the new title (albeit a little wordy?), and I think the focus should have been on the fact that a specific kind of augmentation - semantically similar but structurally different - do well. That is telling me something deeper about model-brain comparisons. I should have pointed this out earlier, but it is good to see that the paper now goes in that direction.
> >
> > I am not very convinced by the explanation for Fig 4. "This suggests that for these networks, either the task objective or training sets (or both) yield weaker alignment with the conceptual representations of visual cortex, allowing for a greater benefit from TTA." Here, for the face network for example, the augmented images are still face images or are they from the other categories? (I didn't find this detail in the methods but please correct me if I missed it. If the data diversity was the issue, I don't see how adding more faces for the face-trained network for TTA would overcome that - but maybe the augmentations were from the other classes, in which case the result would make more sense (please clarify). The other results in the paper also seem more coherent with each other than Fig 4, but it is not something that would affect my rating of the paper.

---

> > > ### Author Response · Authors · 2025-04-21
> > >
> > > Thank you for the question about Figure 4. We would like to clarify that TTA was *not* applied to the images used for pre-training the networks (e.g., face images for the face-trained network). Instead, TTA was applied to the stimuli from the fMRI experiment, which included images from a diverse set of scene categories.

---

### Meta-Review · Area_Chair_DhLY · 2025-05-06

**Ccn Recommendation:** Accept as Proceedings

**Metareview:**

The reviewers raised thoughtful and diverse concerns surrounding the interpretability, methodological clarity, and neuroscientific relevance of the findings. The authors responded comprehensively, conducting new analyses (e.g., dorsal stream ROI, CLIP-based comparisons, null distributions), clarifying conceptual framing (e.g., avoiding overstated claims about semantic abstraction), and addressing figure-specific and methodological questions. The authors’ thorough engagement and the convergence of reviewer consensus indicate that the paper makes a novel and rigorously supported contribution. I therefore strongly recommend acceptance.

**Summary:**

The submission received broadly positive reviews highlighting its conceptual novelty, methodological rigor, and relevance to understanding brain-model similarity. Reviewers appreciated the finding that test-time augmentation (TTA), especially with semantically similar but structurally diverse images, enhances model-brain alignment—a result that challenges common assumptions about model and brain representation alignment. Key concerns included insufficient explanation of why TTA helps, clarity in semantic terminology, interpretation of early versus late layer effects, and dataset limitations (e.g., only 4 NSD subjects). The authors addressed these concerns through additional analyses (e.g., dorsal stream ROIs, CLIP embedding comparisons, empirical null distributions), clarification of methodological choices (e.g., the rationale for 17 augmentations), and revised writing to better articulate implications. Reviewers expressed satisfaction with the revisions, upgrading their evaluation. Overall, the exchange led to a stronger manuscript and consensus on acceptance.

**Expertise:**

2